# DNA Methylation: A Promising Approach in Management of Alzheimer’s Disease and Other Neurodegenerative Disorders

**DOI:** 10.3390/biology11010090

**Published:** 2022-01-07

**Authors:** Gagandeep Kaur, Suraj Singh S. Rathod, Mohammed M. Ghoneim, Sultan Alshehri, Javed Ahmad, Awanish Mishra, Nabil A. Alhakamy

**Affiliations:** 1School of Pharmaceutical Sciences, Lovely Professional University, Phagwara 144411, Punjab, India; sikhsandhu27607@gmail.com (G.K.); surajsinghrathod05@gmail.com (S.S.S.R.); 2Department of Pharmacy Practice, College of Pharmacy, AlMaarefa University, Ad Diriyah 13713, Saudi Arabia; mghoneim@mcst.edu.sa; 3Department of Pharmaceutics, College of Pharmacy, King Saud University, Riyadh 11451, Saudi Arabia; salshehri1@ksu.edu.sa; 4Department of Pharmaceutics, College of Pharmacy, Najran University, Najran 11001, Saudi Arabia; jaahmed@nu.edu.sa; 5Department of Pharmacology and Toxicology, National Institute of Pharmaceutical Education and Research (NIPER)—Guwahati, Changsari, Kamrup 781101, Assam, India; 6Department of Pharmaceutics, Faculty of Pharmacy, King Abdulaziz University, Jeddah 21589, Saudi Arabia; nalhakamy@kau.edu.sa

**Keywords:** epigenetic regulation, DNA methylation, genetic markers, histone modification, Alzheimer’s disease, Parkinson’s disease, Huntington’s disease, amyloid lateral sclerosis

## Abstract

**Simple Summary:**

DNA methylation is an epigenetic modification of genes which affects corresponding gene expression. During the developmental stage, embryonic stem cells undergo various epigenetic modifications to produce different specialized cells. DNA methylation appears as one of the important epigenetic modifications which not only potentiate neuronal development but also have been sought in various neurodegenerative diseases, such as Alzheimer’s disease. The present work focuses on the history of DNA methylation, its role in neurodevelopment functions, and how assessment of DNA hypermethylation and hypomethylation can be utilized for the prognosis of AD and other neurodegenerative diseases. This review also paves the way for the development of novel treatment strategies based on targeting DNA methylation in neurodegenerative diseases.

**Abstract:**

DNA methylation, in the mammalian genome, is an epigenetic modification that involves the transfer of a methyl group on the C5 position of cytosine to derive 5-methylcytosine. The role of DNA methylation in the development of the nervous system and the progression of neurodegenerative diseases such as Alzheimer’s disease has been an interesting research area. Furthermore, mutations altering DNA methylation affect neurodevelopmental functions and may cause the progression of several neurodegenerative diseases. Epigenetic modifications in neurodegenerative diseases are widely studied in different populations to uncover the plausible mechanisms contributing to the development and progression of the disease and detect novel biomarkers for early prognosis and future pharmacotherapeutic targets. In this manuscript, we summarize the association of DNA methylation with the pathogenesis of the most common neurodegenerative diseases, such as, Alzheimer’s disease, Parkinson’s disease, Huntington diseases, and amyotrophic lateral sclerosis, and discuss the potential of DNA methylation as a potential biomarker and therapeutic tool for neurogenerative diseases.

## 1. Introduction

Epigenetic modification refers to heritable changes in gene expression that are not encoded by the DNA sequence [1]. DNA methylation is an inherent epigenetic process in which DNA methyltransferases move a CH_3_ group covalently to the C-5 of the cytosine ring of DNA [2]. DNA methylation occurs at the cytosine in mammalian genetic material. In embryonic cells, around 25 percent of methylation has been identified in a non-CpG context.

Even though the brain contains a higher level of DNA methylation than any other tissue, a low level of 5mC in human genomes is recorded. In general, the methylation of DNA is mostly found at the CpG dinucleotide; however, in somatic cells, the methylation of DNA is highly prevalent on the non-CpG dinucleotide framework. Non-CpG methylation is highly enriched in brain tissue and embryonic stem cells, due to methylation taking place in a non-CpG context [3]. During zygote development, DNA methylation is normally detached and then again established in the embryo approximately during the time of installation [4]. Although several researchers hypothesized that DNA methylation could affect gene function, it was not until the 1980s that researchers showed that methylation in DNA has an impact on gene function and mitosis. This epigenetic modification is now generally accepted as a significant epigenetic factor affecting gene activities [5,6].

DNA methyltransferases (DNMTs) catalyze DNA methylation by transferring a methyl group from S-adenyl methionine (SAM) to the fifth carbon of a cytosine residue, resulting in 5mC. Whenever replication of DNA happens, DNA methyltransferase-1 retrieves the DNA methylation prototype from the parent strands of DNA and transfers it to the newly synthesized daughter strand [7,8]. Although postmitotic neurons in full-grown mammalian brains still demonstrate high levels of DNA methyltransferases, this finding indicates that DNA methyltransferases and epigenetic modification such as methylation in DNA may participate in new functions in the brain [9].

This is more common in species with heavy DNA methylation patterns. However, there can also be no repeat methylation, as investigated in the case of the invertebrate chordate *Ciona intestinalis* [10]. Despite its long evolutionary history, gene methylation is still inadequately understood. Epigenetic regulation also helps to mitigate chromatin disturbance induced by elongating RNA polymerase such as nucleosome displacement [11]. Cytosine-guanine dinucleotides in mammals are methylated on cytosine residues, but cytosine-guanine dinucleotides in promoters are largely unmethylated. Defects in DNA methylation become the cause of many diseases. An increased or decreased level of methylation can result in gene instability [7]. A heritable mechanism for controlling gene expression is a covalent alteration in the genome and proteins present in histone, which is mainly present in chromatin. Histone ends are exposed to an amount of covalent alteration such as acetylation, methylation, phosphorylation, ubiquitination, and sumoylation, all of which control key cellular processes including gene transcription, replication, as well as repair [12]. The addition of three methyl groups in histone has been planned as a condition following DNA methylation, which can be due to interactions between the parts of these histone methylation systems [7]. Histone lysine methyltransferases, suppressor of variegation 3-9 homolog-1 and enhancer of Zeste 2 polycomb repressive complex 2 subunits all perform together and promote their interaction with target promoters [13,14].

CpG islands are stretch of DNA with 500–1500 base pairs with elevated 5′-C-phosphate-G-3′ density compared to other parts of the gene but are mostly unmethylated. CpG islands are home to many gene promoters [15]. CpG islands seem to have evolved to facilitate transcription by controlling the chromatin structure and transcription factor binding. Nucleosomes are small, packaged sections of DNA that are wrapped around histone proteins regularly. DNA becomes less permissive for gene expression as it becomes more closely aligned with histone proteins. CpG islands have fewer nucleosomes than other DNA stretches, which is one of their most distinguishing characteristics [16,17]. Even though 50% of CpG islands have established transcription start sites, they are frequently vacant of common promoter essentials such as TATA boxes [18].

## 2. History and Development of DNA Methylation

DNA methylation, mainly in the cytosine and adenine positions, is responsible for the chromatin structure and dynamics. Out of all epigenetic modifications, methylation, thiouridylation, and pseudo-uridylation of bases in rRNAs and tRNAs are essential for survival. The importance of these modifications is in maintaining double-helical pairing in DNA and preventing mutagenic effects of base modification. They also allow RNA to recall biochemical diversity which is required for their role by protecting genetic material in an unmodified state. Only a small set of DNA alterations have entered evolution and been used to determine biological functions [19].

Mammalian DNA methylation was found almost as soon as DNA was identified as a source of genetic material [20]. Gerlach et al., 1965, reported that modified cytosine was identified in the calf thymus [21]. He proposed that such a proportion was 5-methylcytosine (5mC), since it segregated out cytosine in the same way as thymine segregated from uracil. He also believed that this modified cytosine occurred spontaneously in DNA. Even though many researchers hypothesized that methylation of DNA may affect the expression of genes, it was not until the 1980s that multiple studies showed that methylation of DNA was involved in the regulation of gene and cell division. DNA methylation is now widely recognized as an important epigenetic process influencing gene activity when combined with other promoters. [22].

A number of in silico methods have been developed to determine DNA modification according to the number of enzymes. X-ray crystallography and biochemical studies were used to identify such changes [23]. 5hmC and 5-hydroxymethyluracil synthases along with other enzymes, i.e., DNA base glycosyltransferases, alpha-glucosyltransferase, beta-glucosyltransferase, protein hydroxylases, and histone demethylases, modify bases in DNA and are responsible for moiety transfer from one DNA to another. All DNA methylases belong to monophyletic assemblage, and mainly contain bases in the nucleic acid or amino acid side chain. An examination of the origins of DNA N6A methylases (and related N4C methylases) and their predecessors overlooked their existence in eukaryotes [19].

## 3. Methylation Detection Method

Sodium bisulfite converting and sequencing, the cleavage of DNA by different enzymes, and methylated DNA capturing affinity are the methods used to identify DNA methylation. Methylated DNA immunoprecipitation (Me-DIP), which utilizes a DNA methyl-specific antibody, and methyl encapsulate, which uses methyl-CpG binding domain (MBD) proteins, are the two most widely reported DNA affinity capture methods [24].

The high-performance liquid chromatography-ultraviolet (HPLC-UV) technique, which helps in determining the concentration of deoxycytidine (dC) and methylated cytosines (5mC) contained in a hydrolyzed sample of DNA, is still used today. However, the method’s utility is restricted by the necessity for specialist laboratory apparatus and the need for relatively large amounts of DNA material (3–10 g) to be analyzed [25].

Liquid chromatography coupled with tandem mass spectrometry (LC-MS/MS) is another method used to determine methylation status in DNA. In LC-MS/MS, a small number of samples are required. LC-MS/MS has been verified for identifying methylation levels ranging from 0.05% to 10% in mammalian DNA, and it can confidently identify variations among samples. It can even identify samples which have about 0.25 percent of the total cytosine residues, which equates to 5% differences in global DNA methylation [26].

Because ELISA-based assays have a high risk of considerable changes, they are only useful for estimating DNA methylation in a crude manner. Nonetheless, they are quick and simple to use approaches that work well for detecting substantial changes in global DNA methylation [25]. LINE-1 methylation levels can also be determined using a method that involves bisulfite conversion of DNA followed by PCR amplification of LINE-1 conservative regions. Pyrosequencing is used to determine the methylation status of the amplified fragments, which can resolve discrepancies among samples of DNA as small as 5 percent. Even though this method only focusses on LINE-1 components and hence only on lean CpG sites, it has been proven to accurately reflect overall DNA methylation modification. The method is particularly well-adapted to the examination of high-throughput cancer samples because hypomethylation is frequently linked to a poor prognosis. This approach works best with human DNA, although there are also variations that work with rat and mouse genomes [27].

Differently methylated DNA can also be detected by amplification fragment length polymorphism (AFLP) and restriction fragment length polymorphism (RFLP). These methods have now been replaced by other powerful and more accurate methods. AFLP and RFLP are inexpensive and quickly access the methylation alteration in DNA samples [28].

The luminometric methylation assay is a technique which combines two DNA restriction digest operations that are run in parallel, followed by pyrosequencing to fill in the gaps between the digested DNA strands’ projecting ends. The CpG-methylation-sensitive enzyme HpaII is used in one digesting step, whereas the methylation-insensitive enzyme MspI is used in the other, cutting at all CCGG sites [29]. Bisulfite sequencing is the “gold standard” technology in DNA methylation research. Recent DNA sequencing technology cannot tell the difference between methyl-cytosine and cytosine. The deamination of cytosine into uracil is mediated by bisulfite treatment of DNA, and these transformed residues are read as thymine by PCR amplification and subsequent Sanger sequencing analysis. Five methylated cytosine residues, on the other hand, are independent of this change and remain as cytosine. By comparing the “Sanger sequencing” reads from DNA samples that remain untreated to the cloned sample after bisulfite treatment, the 5mC can be detected. This approach may now be expanded to DNA methylation analysis over a complete genome because of the advent of next-generation sequencing (NGS) technology [30]. Apart from this array or bead hybridization [31], methylation-specific PCR [32], bead array [33], and pyrosequencing can be used to determine the methylation status of the gene of interest [25].

## 4. DNA Methylation in Premature and Mature Brain

The precise timing of de novo methylation and de-methylation in the developing brain is crucial. Multipotent neural progenitor cells (NPCs) go through neurogenesis and astrogliogenesis [34]. The addition and removal of a methyl group in the promoter gene follow the change in neural progenitor cells from neurogenesis to astrogliogenesis and neuron proliferation and development in the adult brain. [35]. The DNMT family of enzymes includes DNMT1 and DNMT3A/DNMT3B enzymes. DNMT1 preferentially leads to methylation of hemi-methylated DNA and further maintains DNA methylation after the replication of DNA, whilst DNMT3A and DNMT3B cause methylation on non-methylated and hemi-methylated DNA equally and are considered as de novo methyltransferase. DNMTs go through significant conformational changes, are capable of oligomerization, and can self-inhibit, all of which could help to regulate their activity [36].

A double knockout study (lacking both DNMT1 and DNMT3A) revealed their importance in the regulation of synaptic plasticity and memory formation through maintaining DNA methylation in neurons [37]. Furthermore, aberrant *bdnf* gene expression (possibly through epigenetic modifications) has been observed in several neurological disorders; however, the inhibition of DNMT resulted in blocking, which was reported to alter *bdnf* DNA methylation status in the hippocampus [38,39] and modulate learning and memory [40,41].

DNMT1 in neural progenitor cells are critical for maintaining the methylated sequence on the glial fibrillary acidic protein promoter throughout mitosis. During the early embryonic stage, *Gfap* gene is methylated; however, at a later stage, it gets demethylated [42]. Methyl CpG binding protein 2 is present in the brainstem and thalamus, which are the brain’s oldest areas. Methyl CpG binding protein 2 interacts with several protein synthesis inhibitors, including DNMTs, and regulates gene expression [43,44,45]. Methyl CpG binding protein 2 is a protein that plays a role in neuronal development, and its loss of phosphorylation causes an abnormal dendritic arborescent figure, a synaptic role, and smoothness [46,47]. The schematic representation of DNA methylation and demethylation is illustrated in Figure 1.

Methyl-binding proteins are another type of protein that works with DNA methylation to control transcription in the CNS [48]. Neuronal activity causes phosphorylation of MeCP2, which leads to changes in gene expression. Synapse development, synaptic flexibility, and learning and recall activity are all impaired when MeCP2 phosphorylation is inhibited. Since phosphorylation is typically a transient alteration, activity-dependent phosphorylation can, temporarily, release methyl CpG binding protein-2 from the enhancer, allowing demethylation of the genome pattern. Methyl group addition or deletion may be to blame for long-term changes in genome number, which is responsible for controlling synapse flexibility, learning, and memory [49,50,51]. The brain is mostly made up of post-mitotic neurons and glial cells that have small proliferative capacities. DNMT1 and DNMT3A are both asserted by mature neurons. In both development and illness, DNMT3B is required for the dynamic programming of epigenetic control. The amino acid sequences of DNMT3A and DNMT3B are very similar. Immunodeficiency, centromeric instability, and facial abnormalities syndrome are caused by DNMT3B mutations [52]. This unexpected finding prompted researchers to investigate further into the function of active gene methylation in adult brain post-mitotic neurons.

## 5. Role of DNA Methylation in Neurological Disorders

Alteration in the methylation pattern of certain genes can modulate neuronal survival and regeneration, which in turn leads to the progression of neuronal degeneration. A summary of recent evidence supporting the hyper/hypomethylation of genes in various neurodegenerative diseases is elaborated in Table 1 and illustrated in Figure 2.

### 5.1. Alzheimer’s Disease

Dementia is a progressive age-related neurodegenerative disease described by progressive cognitive impairment that affects 35.6 million people in the world and is becoming a more pressing issue as the population ages [79]. Extracellular amyloid plaques and intracellular neurofibrillary tangles are hallmarks of Alzheimer’s disease (AD). β-secretase and γ-secretase cleave amyloid precursor protein (APP) sequentially, generating amyloidogenic Aβ peptides that accumulate in the outer space of the cell to form insoluble Aβ plaques.

Decreased global DNMT1 and 5mC in the temporal cortex and reduced global 5mC and 5hmC in the hippocampus of AD patients have been reported [80]. In contrast, some studies support the elevated level of 5mC and 5hmC in the frontal lobe, temporal cortex, and hippocampus in AD [81,82]. Furthermore, reduced 5mC levels in APP, PSEN1, and SERT1 promotors in the brain and blood of AD patients have been recorded [83,84]. Meanwhile, a recent study supported a positive correlation between 5mC and 5hmC values in AD patients [85]. This could be helpful in the differential diagnosis of AD patients from PD (with lower 5hmC level and unchanged DNMT3A expression). The status of methylation and demethylation of various genes in the brains and blood of Alzheimer’s patients is illustrated in Figure 3.

Important variations in DNA methylation profiles of the APP, microtubule-associated protein tau, and GSK3B were also discovered, along with PSEN1, beta-secretase 1 precursor, or apolipoprotein E. Many studies have reported genetic variants linked to enhanced AD susceptibility, but do not limit many other promoters. Almost 28 gene locations are linked with AD, and while little is known about bridging integrator-1 function in AD pathogenesis, it may have a considerable impact on tau pathology, amyloid precursor protein endocytosis, and inflammation in neurons [86,87,88,89].

Most of the AD studies performed to date have mostly considered a gene-directed analysis; thus, the methylation of promoter genes of AD (especially APP) has been widely explored. As mutant APP genes pose a risk factor for AD, it is apparent that epigenetic modifications of the APP promoter leading to enhanced gene expression are also risk factors for AD. West and colleagues reported hypomethylation of the APP promoter in AD patients [90]. Meanwhile, some contrasting studies rejected these findings, suggesting this as an outcome of larger epigenetic modification rather than a specific alteration in methylation [83].

Furthermore, the expression of cortical PSEN1 appeared stable during embryonic development, while upregulated levels have been reported in AD patients. The cortical PSEN1 expression in rodents appears to be tightly regulated by histone modification at various stages of neuronal development. Generally, the PSEN1 gene remains partly methylated and suppressed after development, while the hypomethylation of PSEN1 is reportedly associated with its elevated expression in the AD population [91].

In DNA, when hydrogen is replaced with a hydroxyl group, it is also related to regulating gene expression, although its mechanism is still unclear. In comparison to the moderately constant tissue distribution of 5mC, 5hmC has a wide range of tissue distribution. This is common in the brain, with the cerebral cortex having the largest proportion of 5hmC compared to other parts of the brain [92]. The thromboxane A2 receptor and Sorbin SH3 domain containing 3 (Sorbs3) genes were found to be highly methylated in Alzheimer’s disease. The thromboxane A2 receptor regulates protein synthesis factors such as cyclic adenosine monophosphate response-element binding protein (CREB), which are concerned with neuronal plasticity, protection, and long-term memory development. In the temporal cortex of Alzheimer’s patients, Sorbin SH3 domain containing 3 was found to be highly methylated [93].

This reduction in the methylation level of CREB-regulated transcription factor 1 was found in the hippocampus of AD patients when compared with controls, and this methylation within CREB-regulated transcription factor 1 was inversely connected with phosphorylation in tau expression [94]. Inflammation of neurons is linked to AD. Evidence also revealed that interleukin-1 and interleukin-6 expression levels increase in Alzheimer’s disease, but in the late stages of Alzheimer’s, their levels return to normal [95].

It is also understood that the peripheral blood is also an indicator for AD. Brain-derived neurotrophic factor is responsible for controlling neuronal endurance, division, and flexibility [96,97]. Brain-derived neurotrophic factor is also a key factor in AD. Its level is higher in the peripheral blood of AD patients than in normal controls. Importantly, the percent methylation of specific CpG sites within the brain-derived neurotrophic factor promoter suggests that methylation of the brain-derived neurotrophic factor promoter is associated with clinical manifestations of AD [98,99].

One study also suggested a role for gene expression in AD. H3K4me3 and H3K27ac are responsible for gene alteration. Increased expression of genes causes alterations in immune responses, while decreased expression of genes causes impairment in the learning process. The histone marker H4K16ac causes an alteration in the chromosome which connects it with DNA damage and aging in neurodegenerative diseases. The amount of H4K16a in the aged brain decreases due to the aged brain’s inability to regulate it, which results in changes in brain function [100,101]. An in vitro study on AD found that methylation levels in the 2′,5′-oligoadenylate (2-5A) synthetase gene were reduced [102].

Changes in the methylation levels of SORL1, ABCA7, HLA-DRB5, SLC24A4, and BIN1 also contribute to AD [103]. Hypermethylation of the UQCRC1 gene is responsible for inflammation and oxidative stress in AD [104]. An increase in DNA methylation in TREM2 was also reported, which causes alteration in AD biomarker TREM2 mRNA expression [105]. Hyper methylation of OPRM1 and OPRL1 genes in AD patients has also been reported in many studies that demonstrate the role of opioid receptors in the diagnosis of AD [106]. A decrease in the methylation level of the PICALM gene leads to an alteration in the cognitive behavior of AD patients [107]. The ANK1 gene’s elevated methylation profile demonstrated its role in AD [108]. OTX gene methylation changes also lead to AD [101].

Apart from this, DNA hydroxymethylation is also one of the epigenetic mechanisms that lead to the progression of the disease. As discussed above, hypermethylation of the ANK1 gene leads to AD. Hypo-hydroxymethylation of the ANK1 gene is also one of the factors related to AD. Hypermethylation of the PLD3 gene also plays a role in AD [109]. In the 5xFAD mouse model, as well as in the 3xTg-AD mouse model, changes in DNA methylation and hydroxymethylation patterns in genes were observed. These models proved that a reduction in hydroxymethylation also plays an important role in AD [110]. Hydroxy methylation of genes is also related to cognitive behavior and memory impairment. Hydroxy methylation of the DNA sequence gives rise to 5-hydroxymethycytosine, which is present in large amounts in the brain and plays a role in neurodevelopment. DNA hydroxymethylation in the TREM2 gene was found to be increased in AD patients. Different hydroxymethylated regions can be the reason for neuritic plaques and neurofibrillary tangles. Alterations in DNA patterns in the prefrontal cortex cause neurogenesis [111,112]. Levels of 5-hydroxymethylcytosine were also found to be elevated in mitochondria, which also suggested its role in AD [113].

### 5.2. Parkinson Disease

In the elderly population, Parkinson’s disease (PD) is the most widespread disease caused by the degeneration of neurons, mainly dopaminergic neurons. In post-mortem brains, Lewy bodies, irregular protein aggregates contained inside nerve cells, and gradual depletion of dopamine neurons present in the substantia nigra have been discovered [114]. In the last few years, complete hereditary screening of Parkinson’s disease families has sought to detect mutations linked to the disease, which will provide a better understanding of the actual mechanisms involved in the disease. Several gene locations that are associated with familial Parkinson’s disease, such as Parkinson’s disease (PARK 1-15) and other genes, have been identified through genetic studies. Other genes linked to sporadic PD have been identified, including leucine-rich repeat kinase-2, alpha-synuclein, microtubule-associated protein tau, and encoding for the lysosomal enzyme glucocerebrosidase [115,116].

Alpha-synuclein aggregation leads to Lewy body expansion, a characteristic of PD [117,118]. DNA methylation has been projected as a possible means for the deregulation of alpha-synuclein in PD [119]. The use of a DNA methylation inhibitor reduced CpG-2 methylation while significantly raising alpha-synuclein mRNA and protein levels. The addition of a methyl group, at least at the intrinsic CpG-2 island, has been shown in recent studies to regulate alpha-synuclein gene action [120].

One study also claimed that alterations in methyl group addition levels in alpha-synuclein take place in many parts of the brain. Both the addition and removal of methyl groups were found in the promoter gene region as well as in intron 1 of different Lewy body disease/PD levels [121]. Other studies would be required to overcome these doubts about the validity of DNA methyl group removal at the alpha-synuclein intron 1 concerning PD, given the inconsistencies. So, this DNA methylation modification may act as a biomarker for Lewy body-related diseases rather than a particular biomarker for PD [122,123].

Apart from SNCA, beta-synuclein (SNCB) also plays a major role in PD. In vitro, SNCB prevents the formation of alpha-synuclein fibril aggregation, suggesting that it may help in protecting neurons that are prone to degeneration [124]. The promoter of the beta-synuclein gene was originally found to be unmethylated in the brain. Bisulfite sequencing of the beta-synuclein promoter revealed no 5mC along the cytosine phosphate guanine island in four uncontaminated disperse Lewy body pathology cases [125].

Peptidyl arginine aminases (PADs) also play a role in PD, and their promoter is found to have a reduced methylation level in the brain, but it is the opposite in white matter. Other than alpha-synuclein, some genes such as ubiquitin carboxyl-terminal hydrolase isozyme L1 (UCHL1) promoter, ATP13A2 promoter, Parkin (PARK2) gene, and other clock genes such as period circadian regulator (PER1), period circadian regulator-2, cryptochrome circadian regulator-1, cryptochrome circadian Regulator-2, neuronal PAS domain protein 2 (NPAS2), and brain and muscle ARNT-like 1 (BMAL1) have also been measured in genomic DNA isolated from PD patients to check their role in PD. Mutations in the ATP13A2 promoter and PARK2 are related to PD. However, in the case of the UCHL1 promoter, ATP13A2 promoter, and PARK2, there were no variations in 5′-C-phosphate-G-3′ methylation percentages between PD cases and control groups [126,127,128,129,130]. DNA methylation was found in the cryptochrome circadian regulator 1 and neuronal PAS domain protein 2 promoters, but not in other gene promoters studied [131].

The microtubule-associated protein tau (MAPT) gene is associated genetically with PD. In MAPT, the H1 haplotype had more DNA methylation than the H2 haplotype (this haplotype was linked appreciably with PD). The MAPT gene was found to have increased methyl group levels in the cerebellum, but not in the putamen of PD patients [132,133,134].

In the CpG-1 and CpG-2 islands of PARK7, there is no methylation in both the PD and the control groups, where the PGC-1 promoter is highly methylated [135,136]. When investigated, single CpG sites of both Fanconi anemia complementation group C (FANCC) and tankyrase 2 (TNKS2) showed differences in methylation patterns in the PD and control groups. Another study revealed that about 20 genes were found to have DNA methylation differences in PD patients [137].

A comparative analysis of DNA methylation studies in the brain and blood of PD patients suggested alteration in their DNA expression. KCTD5, VAV2, MOG, TRIM10, HLA-DQA1, ARHGEF10, GFPT2, HLA-DRB5, TMEM9, MRI1, MAPT, HLA-DRB6, LASS3, GSTTP2, GSTTP were found to be hypermethylated, and DNAJA3, JAKMIP3, FRK, LRRC27, DMBX1, LGALS7, FOXK1, APBA1, MAGI2, SLC25A24, GSTT1, MYOM2, ME 886, TUBA3E, TMCO3 genes were hypomethylated in the brains and blood of PD patients [138].

Recent evidence elaborating DNA methylation of different genes in AD and PD is presented mentioned in Table 2.

### 5.3. Huntington Disease

Huntington’s disease (HD) is caused by a CAG repeat mutation in the huntingtin gene. Huntingtin is a disease protein that has been established to affect a variety of epigenetic markers, such as histone modifications, and mainly DNA methylation [155]. Huntingtin (Htt), a protein with an enlarged glutamine domain, is prearranged by the mutant Htt gene. The specific mechanism by which the mutant huntingtin protein causes degeneration of neurons is unidentified [156]. Many interactions with certain transcription factors are involved, such as meddling with protein synthesis mechanisms and modifications of histones after protein synthesis. The mutant Htt has a large influence on gene expression, by pushing chromatin towards a thicker form [157,158]. Changes in DNA methylation have been associated with Htt protein expression in a variety of HD model systems, and in the human HD brain, according to numerous studies. Cedric Ng and his coworkers revealed differential methylation in DNA patterns in STHdhQ111 cells compared to WTQ7 cells of mice by using condensed representation bisulfite sequencing to check DNA methylation [77]. Transgenic mice had decreased levels of 5-hydroxymethylcytosine in the striatum and brain compared to controls [159]. One study claimed that the hairy enhancer of split-4 (HES4) gene promoter has a major site-specific DNA methylation modification in HD patients [160].

In the putamen of HD patients and the striatum of R6/1 and R6/2 mice, 5-methylcytidine-5′-monophosphate and 5-hydroxymethylcytosine contents were detected in the 5′ untranslated region (5′UTR) region of the adenosine receptor gene-A2A (ADORA2A). The pathological drop in adenosine A2a receptor expression levels seen in HD is linked to the abnormal methylation patterns of the adenosine receptor gene A2a [161]. Growth arrest and DNA-damage-45a (Gadd45a) expression were shown to be lower in the striatum of animals and muscle of transgenic mice, while growth arrest and the expression of DNA-damage-45g, a member of a group of genes whose transcript levels are increased following stressful growth arrest conditions, were found to be elevated in HD STHdhQ111 cells. Finally, one more investigation revealed that ring finger protein 4, another gene associated with DNA demethylation, was expressed differently in HD mice [162].

One study also suggested that altered DNA methylation profiles of CLDN16, DDC, and NXT2 also play a role in the progression of the disease [163]. Twelve genes—FBXL5, S100P, PRDX1, COPS7B, SP1, SEC24C, PDIA6, USP5, GRAP, POP5, WRB, and PCSK7—were also found to be differently methylated in HD patients’ blood. Sox2, Pax6, and Nes genes were found to have increased methylation levels, which tends to reduce their expression and result in impaired neurogenesis [73,164].

### 5.4. Amyotrophic Lateral Sclerosis

Amyotrophic lateral sclerosis (ALS) is a neurodegenerative disease. Motor neurons in the motor cortex, brain stem, and spinal cord degenerate in this condition. Oxidative stress, glutamate excitotoxicity, impaired axonal transport, neurotrophic deprivation, neuroinflammation, apoptosis, altered protein turnover, and mitochondrial dysfunction are the mechanisms involved in the pathogenesis of ALS. Changes in the immune system, more physical activity, exposure to toxins, and dietary factors can also lead to the progression of this disease [165]. Alteration of DNA methylation sequences also plays a role in ALS. Elevated levels of 5-methylcytosine participate in disease pathology. This is due to the increased activity of DNMTs [166].

Increased levels of 5-hydroxymethylcytocise lead to aging and a rise in oxidative stress [167]. Methylation changes influence immune-related genes, i.e., TREM2, chemokine (C-C motif) receptor1/RANTES receptor (CCR1), SLC11A1, the transmembrane receptor C-type lectin domain family 4 member A isoform 1 (CLEC4A), and the IgE receptor (FCER1G). All these genes are elevated in ALS and cause an increase in the number of immune cells [168,169].

STAT5A and C/EBPB are transcription factors that activate other genes such as interleukin 6, which are responsible for the pathogenesis of ALS and neurodegeneration [170]. Changes in gene expression of Slit-Robo Rho GTPase activating protein 1 (SRGAP1), Crumbs homolog 1 (CRB1), MSX2, MLC1, CTNND2, AXL, RUNX3, NNAT, and NRN1 cause neurodegeneration. Crumbs homolog 1 was found to be hypomethylated and responsible for intellectual disability and neurodegeneration in ALS patients [171,172,173]. Scientists also reported a modest overlap of four concordant epigeneses; Purkinje cell protein 4 (PCP4), catenin (CTNNAL1), fibroblast growth factor 18 (FGF18), and flavin-containing monooxygenase 1 (FMO1) [164]. Mutations in SOD1, FUS, TARDBP, and C9orf72 cause oxidative stress in ALS [173].

Mitochondria are the powerhouses of the cell. Alterations in mitochondria can also be related to ALS. Mito-epigenetics assesses the DNA methylation and DNA hydroxymethylation levels in mitochondria. One study also proved the role of mitochondria in ALS by demonstrating that mitochondrial DNA methylation patterns are altered in the skeletal muscles and spinal cord of ALS patients [173]. D-loop hypomethylation in SOD1 carriers could be related to a rise in mtDNA methylation, which gives rise to an increase in mtDNA methylation that causes an increase in oxidative stress. SODI is an antioxidant enzyme. A mutation in this enzyme causes ALS disease.

## 6. Targeting DNA Methylation in Management of AD and Other Neurodegenerative Diseases

Identifying DNA methylation in peripheral blood and brain samples could be a promising biomarker for diagnosing AD. In this regard, the hypermethylated *APP* gene appears as a promising biomarker for AD prognosis. Based upon these observations, several researchers have carried out some preliminary studies which support the assertion that targeting DNMT could be beneficial in halting amyloid pathology and other neurodegenerative diseases. Some of the corresponding studies are presented in Table 3 and are elaborated herewith.

Epigallocatechin gallate, epigallocatechin 3-gallate, tea catechin, and catechin derivatives are example DNMTs inhibitors that have proven their role in the treatment of AD. Epigallocatechin-3-gallate (EGCG) prevents misfolded proteins from undergoing fibrillation and protects from cell death in Aβ-treated neurons. Etanercept also helps to treat AD by modulating the immune system. Neurodegeneration medications include 5-aza-2′-deoxycytidine (decitabine) and 5-azacytidine (azacitidine), as well as the small molecules hydralazine and procainamide [174].

Wang et al., 2013, tested 5-aza-2′-deoxycytidine (5-aza-dC), a DNMT inhibitor, in the treatment of Parkinson’s disease. The scientist reported that 5-aza-dC induced CpG demethylation in the promoter and upregulated transcriptional levels of the α-synuclein gene. 5-aza-dC has been demonstrated to increase the expression of tyrosine hydroxylase dopamine production and alpha-synuclein expression. [175]. In general, if levodopa is shown to work via an epigenetic pathway, existing treatments should be reassessed to elucidate fresh epigenetic characteristics and develop innovative and more targeted drugs [178]. Vitamin B, folic acid, and SAMe are the main methylation storage compounds submitted to clinical trials for the treatment of neurodegenerative diseases [174]. Folate and vitamin B6 are also prescribed by a doctor to treat elevated levels of homocysteine, which is a known risk factor for AD [174].

Pan et al., 2016, proved that decitabine and FdCyd, DNMT inhibitors, can attenuate neurotoxicity in HD patients. DNMT inhibition leads to the restoration of the expression of Bndf and can be used as a therapeutic target for treatment [176].

5-aza-cytidine (5-azaC), 5-aza-2-deoxycytidine (5-azadC, decitabine), zebularine, and RG108 are drugs that inhibit DNA methylation and can be used in the improvement of ALS. RG108 blocks the methylation in motor neurons and causes improvement in disease [177].

## 7. Conclusions

This review summarizes those epigenetic modifications that are responsible for many genes’ functions in the body. Studies have investigated the relationship between DNA methylation of genes and their level in the pathology of neurodegenerative diseases. DNA methylation influences the pathophysiology of age-related disease, aging, and dementia [179,180], Modifications in chromatin organization, transcriptional changes, and a variety of neurological illnesses and diseases are all linked to abnormal methylation changes [181]. In AD, the DNA methylation of genes such as amyloid precursor protein, PSEN1, MAPT, apolipoprotein E, presenilin-1, beta-secretase 1 precursor, or apolipoprotein E, Sorbin SH3 domain containing 3 (Sorbs3) and BDNF was found to be altered in three parts of the brain. Based on the above-mentioned points, it may be proposed the identification of DNA methylation in peripheral blood and brain samples could provide valuable insights for AD diagnosis. Furthermore, designing suitable DNA methylation promotors or inhibitors could provide a novel target for the management of AD and other neurodegenerative diseases.

## Figures and Tables

**Figure 1 biology-11-00090-f001:**
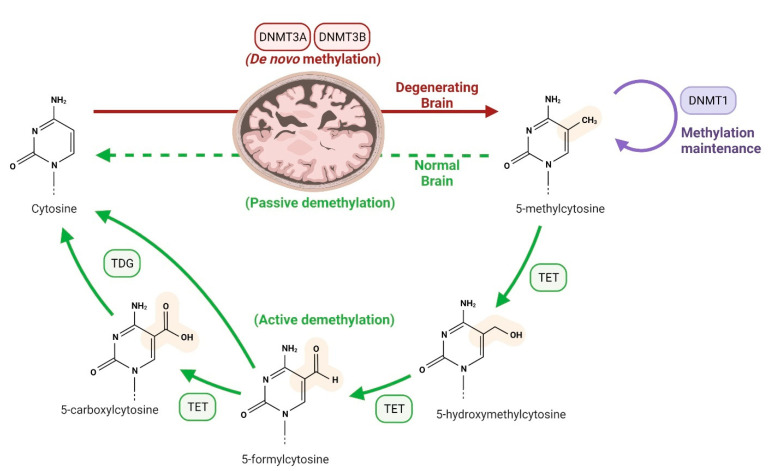
A systemic representation of DNA methylation and demethylation. By means of the action of DNA methyltransferases, the methyl group is transferred to the 5th carbon in cytosine. Generally, DNA methylation is initiated by DNMT3A/3B (de novo pathway) and is maintained by DNMT1. Most gene expression is suppressed by DNA methylation, which contributes to the development of neurodegeneration. On the other hand, methylated cytosine turned into cytosine through an active/passive demethylation process. This cycle regulates the gene expression, while under certain environmental conditions, abnormality in this cycle may contribute to the development of neurodegenerative diseases.

**Figure 2 biology-11-00090-f002:**
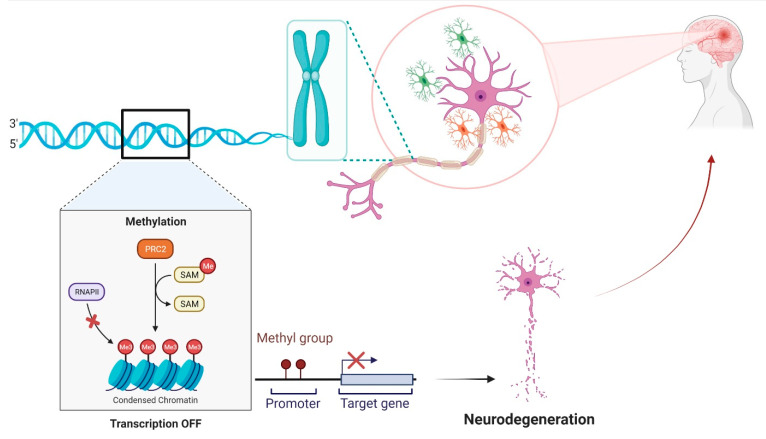
Schematic representation of DNA methylation causing the inhibition of target gene expression and progression towards neurodegeneration.

**Figure 3 biology-11-00090-f003:**
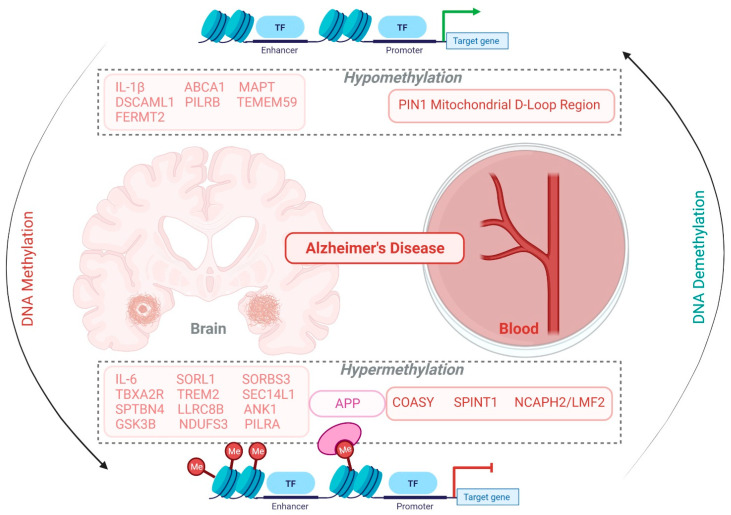
Differential methylation pattern of various genes in brain and peripheral blood in Alzheimer’s disease. This image illustrates the alteration in the expression of different genes in the AD brain and peripheral blood circulation due to the hyper/hypomethylation of DNA. The genes encoded in light red are specific to the brain region, while the genes encoded in red color are restricted to peripheral blood circulation. In addition, hypermethylation of the APP gene is reported as a common gene in the AD brain and peripheral blood circulation.

**Table 1 biology-11-00090-t001:** DNA methylation status of different genes in neurodegenerative diseases.

Disease	Sample	Methylation (Hyper/Hypo)	Experimental Method	Gene	Ref.
AD	Blood	Hypermethylation	Bisulphite sequencing PCR and methylation-specific PCR are used	SIRT1	[53]
AD	Dorsolateral prefrontal cortex tissue	Differently methylated	CpG sites generated using a bead assay	SORL1, ABCA7, HLA-DRB5, SLC24A4, BIN1.	[54]
AD	Hippocampus	Hypomethylation	Bisulfite cloning sequencing of CpG sites in two promoter regions Prom1 and Prom2	CREB-regulated transcription factor 1	[55]
AD	Blood	Hypermethylation	Bisulfite treated DNA was analyzed by melting curve analysis-methylation assay	UQCRC1	[56]
AD	Hippocampus	Hypermethylation	Bisulfite cloning sequencing and further measured by 5-hydroxymethycytosine (5hmC)	TREM2	[57]
AD	Blood	Hypermethylation	Dual-luciferase assays	OPRM1, OPRL1	[58]
AD	Blood	Hypomethylation	Quantitative bisulfite-PCR pyrosequencing	PICALM	[59]
AD	Brain	Hypermethylation	Bisulfite pro-sequencing	ANK1 gene	[60]
AD	Hippocampus	Hypermethylation	RT-qPCR	PLD3 gene	[61]
PD	Postmortem human brain samples (frontal cortex)	Hypermethylation	Illumina Infinium array	MRI1, TMEM9	[62]
PD	Postmortem human brain samples (frontal cortex)	Hypomethylation	Illumina Infinium array	GSST1, TUBA3E, KCNH1	[62]
PD	Brain tissue	Hypomethylation	Fluorescence-activated nuclei sorting and bisulfite pro-sequencing	CpGs located in SNCA intron 1	[63]
PD	Blood	Differently methylated	Cross-sectional analysis of blood methylation	SRSF7, ADNP, GDNF, SYN3, CPLX1, SNCA, TREM2.	[64]
PD	Blood and saliva	Altered methylation	Illumina Infinium array	ABCB9, C1orf200, AZU1, LARS2, PARK2, LRRK2, APC, AXIN1	[65]
PD	Brain	Differently methylated	Genome wide screening and RNA sequencing	ARFGAP1, DUSP22 promoter, SNCA	[66]
PD	Leukocytes	Hypomethylation	Methylation-specific PCR	NPAS2	[67]
PD	Brain	Hypermethylation	Bisulfite sequencing and micro array gene expression analysis	PGC1-α	[68]
PD	Brain	Hypomethylation	Genome wide methylation	CYP2E1	[69]
PD	Blood	Hypomethylation	-	NOS2	[70]
PD	Leukocytes, Brain	Hypermethylation	Bisulfite pyrosequencing and MAPT promoter methylation assay	MAPT	[71]
PD	Brain	Hypermethylation	Illumina Infinium array	FANCC/TNKS2	[72]
HD	Striatal cells carrying polyglutamine-expanded HTT (STHdhQ111/Q111) and wild-type cells (STHdhQ7/Q7)	Altered DNA methylation	mRNA-Seq, ChIP-Seq assay and Motif Scanning	Htt	[73]
HD	Prefrontal cortex	Differently methylated	Fluorescence-based nuclei sorting (FACS)-ChIP-seq	HES4	[74]
HD	Putamen of HD patients and striatum of mice	Differently methylated	Bisulfite sequencing and TaqMan PCR	ADORA2A	[75]
HD	Blood	Differently methylated	Microarray methylation	CLDN16, NXT2, DDC.	[76]
HD	Blood	Differently methylated	mRNA-Seq, ChIP-Seq assay and motif scanning	FBXL5, S100P, PRDX1, COPS7B, SP1, SEC24C, PDIA6, USP5, GRAP, POP5, WRB, PCSK7.	[77]
ALS	Postmortem spinal cord tissue	Hypomethylation	Bisulfite pyrosequencing, genome-wide expression profiling, and RT-PCR	MLC1, CRB1, CTNND2, FURIN, SLC31A1, CMTM3, STAT5A, SRGAP1, LPXN, PLD4, OBFC2A, TXNIP, PSAP, SLC35E1, RBM38, CLEC4A, HMHA1, PLSCR1, AXL, PHYHD1.	[78]
ALS	Postmortem spinal cord tissue	Hypermethylation	Bisulfite pyrosequencing, genome wide expression profiling, and RT-PCR	LUM, SLC13A4, GJB2, TYRP1, CLDN19, LINGO2, PLEKHA4, NNAT, TSPAN18, PLCB4, TMEM139, PNMAL1, DMBT1, TNFSF10, NNAT, PCP4, MAB21L2, PEG10, TMEM139, KCNJ12, FGF18.	[78]

**Table 2 biology-11-00090-t002:** Recent evidence supporting the role of DNA methylation in Alzheimer’s and Parkinson’s disease.

Methylation of DNA	Gene/Target/Pathway Involved	Effect	Model	Experimental Method	Outcomes	Ref.
5-mC	*B3GALT4, ZADH2*	Decrease	AD and healthy patients	Rey Auditory Verbal Learning Test (RAVLT), Trail Making Test Part B (TMT-B), INNOTEST assays, and Triplex assay	Hypomethylation of *B3GALT4, ZADH2* associated with the level of AB and tau in CSF	[139]
5-mC	*HOXA3, GSTP1, CXXC1-3, BIN1*	Increase	AD and healthy patients	Laser-assisted microdissection and Infinium DNA Methylation 450K analysis	504 DMCs and 237 DMRs were identified and increased in the 5mC pyramidal layer, which is associated with oxidative stress	[140]
5-mC	*KIAA056*	Decrease	NFT pathology stages I-IV	Bisulfite sequencing and Infinium Human Methylation 450 BeadChip	Downregulation of 5mC in *KIAA056* and in NFT pathology cases	[141]
5-mC	*ANKRD30B, ANK1*, Cell adhesion	Increase	AD and neurotypical patients	Genome-wide DNA methylation, mRNA expression profiling, functional enrichment analysis, and differential methylation of genes	856 DMCs were identified along with a correlation between 5-mC and gene expression	[142]
5-mC	*WNT5B, ANK1, ARD5B*	Increase and decrease	AD patients	Illumina Infinium Human Methylation 450K microarray	Increased 5-mC level in *WNT5B, ANK1,* and decreased in *ARD5Bz*	[143]
5-mC	Amyloid neuropathy and neurogenesis	Decrease	AD and healthy patients	RNA sequencing, aging analysis, gene annotation, and enrichment analysis	Identification of 1224 DMRs, enhancement in the *DCSAML1* gene which targets *BACE1*	[144]
5-mC	-	Decrease	AD and healthy patients	Immunohistochemistry	Downregulation of 5-mC and negative correlation between 5mC and amyloid plaque level	[145]
5-mC	-	Increase	AD patients and preclinical samples	Immunohistochemistry	Upregulation of 5-mC and hippocampus gyrus in both clinical and preclinical cases	[146]
5-mC	-	Increase	Early and late-onset AD patients	Immunohistochemistry	Upregulation of 5mC in middle frontal gyrus and middle temporal gyrus in AD patients and shows a positive correlation with AD biomarkers	[147]
5-mC	*AS3MT, WTI, TBX15*	Decrease	AD with psychosis and without psychosis patients	Immunohistochemistry	Decrease level of *AS3MT, WTI, TBX15* gene associated with AD patients	[148]
5-mC	-	Decrease	Early and late AD patients	Immunohistochemistry	Genetic dysregulation may be occurring in astrocytes and NF-positive pyramidal neurons in AD	[149]
IL-1β Promoters	IL-1β	Decrease	BALB/c mice (3–4- and 18–20-month-old)	LPS-induced neuroinflammation and Quantitative PCR (qPCR)	Microglial transferred to M1 phenotype which causes neuroinflammation and neuronal cell damages	[150]
SNCA Promoters	SNCA	Decrease	Healthy and PD patients	qPCR	Aggregation of a-syn, neuronal damage of DA, and neuroinflammation is triggered by activating glial cells	[151]
PGC-1α Promoters	PGC-1α	Increase	Human brain of PD and healthy patients	Bisulfite sequencing, Microarray gene expression analysis, ELISA analysis	Up-regulation of neuroinflammation, ER stress, epigenetic modification, and ROS production	[152]
TNF-α Promoters	TNF-alpha	Decrease	PD and healthy patients	Bisulfite PCR and sequencing	SNpc cells could underlie the increased susceptibility of dopaminergic neurons to TNF-alpha-mediated inflammatory reactions.	[153]
NOS2 Promoters	NOS2	Decrease	PD and healthy patients	Qiagen’s Assay	Down-regulation of NO production to deactivate the microglial	[154]

**Table 3 biology-11-00090-t003:** Recent advances in the management of AD and other neurogenerative diseases by targeting DNA methylation.

Neurodegenerative Disease	Drug	Class of Drug	Inference	Reference
AD	Epigallocatechin gallate, epigallocatechin 3-gallate, tea catechin, tea vigo, catechin deriv.,	DNMT inhibitors	Improve memory, prevent cell death in Aβ-treated neurons, Aβ aggregation.	[174]
AD	Vitamin B6, folate, Folacin; Pteroylglutamic acid	SAMe methyl donors	Attenuate homocystine level	[174]
PD	5-Aza-2′-Deoxycytidine	DNMTs inhibitor	Upregulate tyrosine hydroxylase, dopamine production, and alpha-synuclein expression	[175]
HD	decitabine and FdCyd	DNMTs inhibitors	Restore expression of Bndf	[176]
ALS	RG108	DNMTs inhibitors	Block DNA methylation accumulation in motor neurons	[177]

## Data Availability

No new data were created or analyzed in this study. Data sharing does not apply to this article.

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
