# Peer review of "DNA Methylation: A Promising Approach in Management of Alzheimer’s Disease and Other Neurodegenerative Disorders"

_biology, 2022, doi:10.3390/biology11010090_

Round 1

Reviewer 1 Report

This review is trying to provide a comprehensive summary on the DNA methylation signatures related to neurodegeneration diseases. Generally, the manuscript is well structured and having a clear flow. There’re some comments for the further improvements.

  1. Starting with the main concern, the authors tried gathering all the methylation biomarkers associated to 4 different diseases. However, the lack of methylation detection method used in each reference limits the audiences to know the data quality and sequencing resolution to compare the significance and robustness of their findings. To get a better understanding on these methylation biomarkers, few columns described the experiment methods can be added to table 1 and 2.
  2. Some existed reviews provide good examples discussing on the same topic, it’s better to cite them and let readers to track the development and history of this field, such as:
  3. Delgado-Morales, R. & Esteller, M. Opening up the DNA methylome of dementia. Mol Psychiatry 22, 485-496, doi:10.1038/mp.2016.242 (2017).
  4. Armstrong, M. J., Jin, Y., Allen, E. G. & Jin, P. Diverse and dynamic DNA modifications in brain and diseases. Hum Mol Genet 28, R241-R253, doi:10.1093/hmg/ddz179 (2019).
  5. Salameh, Y., Bejaoui, Y. & El Hajj, N. DNA Methylation Biomarkers in Aging and Age-Related Diseases. Front Genet 11, 171, doi:10.3389/fgene.2020.00171 (2020)
  6. DNA methylation is observed mostly anti-correlated to gene expression, not likely the Figure 3 showed an opposite way between the illustrations and the labels on hyper or hypo.
  7. A couple of minor concerns need additional clarifications and justifications:
  8. Line #18, “transfer of methyl group from 5-methylcytosine to C5 position of cytosine”, is a confused statement, methyl group is transferred from SAM.
  9. Line #29, “therapeutic tool” is better to be replaced by “therapeutic targets”.
  10. Line #39 is the wrong interpretation of the paper cited. Non-CpG methylation is highly enriched in embryonic stem cell and brain tissue, but doesn’t mean methylation mostly occurring at non-CpG context.
  11. Line#45, “gene activities” is better than “gene actions”.
  12. Line#86, it’s hard to understand the sentence end up with “critical (CNS)”.

Author Response

This review is trying to provide a comprehensive summary on the DNA methylation signatures related to neurodegeneration diseases. Generally, the manuscript is well structured and having a clear flow.

Response: Authors would like to appreciate reviewer’s consideration and valuable time and feedback for the improvement of our manuscript. Now we have revised the manuscript thoroughly and changes has been made using track changes.

There’re some comments for the further improvements.

  1. Starting with the main concern, the authors tried gathering all the methylation biomarkers associated to 4 different diseases. However, the lack of methylation detection method used in each reference limits the audiences to know the data quality and sequencing resolution to compare the significance and robustness of their findings. To get a better understanding on these methylation biomarkers, few columns described the experiment methods can be added to table 1 and 2.

Response: As suggested, a separate section for detection methods for DNA methylation has been added (as section 2). Table 1 and 2 has now been updated with experimental methods column.

  1. Some existed reviews provide good examples discussing on the same topic, it’s better to cite them and let readers to track the development and history of this field, such as:

Delgado-Morales, R. & Esteller, M. Opening up the DNA methylome of dementia. Mol Psychiatry 22, 485-496, doi:10.1038/mp.2016.242 (2017).

Armstrong, M. J., Jin, Y., Allen, E. G. & Jin, P. Diverse and dynamic DNA modifications in brain and diseases. Hum Mol Genet 28, R241-R253, doi:10.1093/hmg/ddz179 (2019).

Salameh, Y., Bejaoui, Y. & El Hajj, N. DNA Methylation Biomarkers in Aging and Age-Related Diseases. Front Genet 11, 171, doi:10.3389/fgene.2020.00171 (2020)

Response: As suggested the mentioned references has now been added in the manuscript.

  1. DNA methylation is observed mostly anti-correlated to gene expression, not likely the Figure 3 showed an opposite way between the illustrations and the labels on hyper or hypo.

Response: As suggested the labels for hyper/hypo has been changes in the Figure 3.

  1. A couple of minor concerns need additional clarifications and justifications:
    1. Line #18, “transfer of methyl group from 5-methylcytosine to C5 position of cytosine”, is a confused statement, methyl group is transferred from SAM.

Response: The sentence has been revised for better clarity.

  1. Line #29, “therapeutic tool” is better to be replaced by “therapeutic targets”.

Response: As suggested, “…tool.” Has now been replaced with “… targets”.

  1. Line #39 is the wrong interpretation of the paper cited. Non-CpG methylation is highly enriched in embryonic stem cell and brain tissue, but doesn’t mean methylation mostly occurring at non-CpG context.

Response: The sentence has been revised to provide better clarity.

  1. Line#45, “gene activities” is better than “gene actions”.

Response: As suggested the change has been inserted at appropriate place.

  1. Line#86, it’s hard to understand the sentence end up with “critical (CNS)”.

Response: The ambiguity has been removed to improve the understanding of the sentence.

Reviewer 2 Report

The review DNA Methylation: a promising approach in management of neurodegenrative disorders  discuss the ptential use of DNA methylation as a biomarker for these disorders. The review is interesting but I have some minor concerns:

1.- What happens in these pathologies with global DNA methylation and hydroxymethylation?

2.- The tittle is" a promising approach in management of neurodegenerative disorders" and at the end of the abstract authors say "DNA methylation as a potential biomarker and therapeutic tool for neurodegenerative disease". The review describes and discuss the bibliography about  DNA methylation as a biomarker but not as a therapeutic tool.  There are connection between DNA methylation and the response to treatments? Is DNA methylation useful for the development of new drugs? This issue should be added or the title and abstract must be changed.

Author Response

# Reviewer 2

The review DNA Methylation: a promising approach in management of neurodegenrative disorders discuss the ptential use of DNA methylation as a biomarker for these disorders.

Response: Authors would like to thank reviewer for their valuable comments for the improvement of our manuscript. Now we have revised the manuscript thoroughly and changes has been made using track changes.

The review is interesting but I have some minor concerns:

  1. What happens in these pathologies with global DNA methylation and hydroxymethylation?

Response: The enhancement in global DNA methylation has been observed in late onset AD patients along with enhanced expression of dnmt1 gene. Thus, global methylation in peripheral samples appears as useful diagnostic markers for developing AD (Di Francesco et al., 2015).

The levels of hydroxymethylation cytosine appears higher in neurons of human AD brain (Coppieters et al., 2014; Zhao et al., 2017). However global status of hydroxymethylation is yet to be analysed.

  • Di Francesco A, Arosio B, Falconi A, Micioni Di Bonaventura MV, Karimi M, Mari D, Casati M, Maccarrone M, D'Addario C. Global changes in DNA methylation in Alzheimer's disease peripheral blood mononuclear cells. Brain Behav Immun. 2015 Mar;45:139-44. doi: 10.1016/j.bbi.2014.11.002. Epub 2014 Nov 13. PMID: 25452147.
  • Coppieters N, Dieriks BV, Lill C, Faull RL, Curtis MA, Dragunow M. Global changes in DNA methylation and hydroxymethylation in Alzheimer's disease human brain. Neurobiol Aging. 2014 Jun;35(6):1334-44. doi: 10.1016/j.neurobiolaging.2013.11.031. Epub 2013 Dec 4. PMID: 24387984.
  • Zhao J, Zhu Y, Yang J, Li L, Wu H, De Jager PL, Jin P, Bennett DA. A genome-wide profiling of brain DNA hydroxymethylation in Alzheimer's disease. Alzheimers Dement. 2017 Jun;13(6):674-688. doi: 10.1016/j.jalz.2016.10.004. Epub 2017 Jan 6. PMID: 28089213; PMCID: PMC6675576.
  1. The tittle is" a promising approach in management of neurodegenerative disorders" and at the end of the abstract authors say "DNA methylation as a potential biomarker and therapeutic tool for neurodegenerative disease". The review describes and discuss the bibliography about DNA methylation as a biomarker but not as a therapeutic tool. There are connection between DNA methylation and the response to treatments? Is DNA methylation useful for the development of new drugs? This issue should be added or the title and abstract must be changed.

Response: Authors appreciate keen observation of reviewer, and the suggested changes has been included in the manuscript. To demonstrate, targeting DNA methylation in AD and other neurogenerative diseases has been added as “Section 6 (Targeting DNA methylation in management of AD and other neurodegenerative dis-eases)” and a separate table has been inserted as Table 3.

Reviewer 3 Report

The review by Kaur and colleagues is aimed at showing the role of the DNA methylation in neurodegenerative disorders and adds to a large body of similar papers.
Unfortunately the manuscript results quite superficial and repetitive without drawing any relevant perspectives.
The review needs of an extensive editing of both the English language and the style.
Just to mention some examples:
1) Line 19: better to use "epigenetic modifications" instead of "epigenetic alteration".
2) Line 20-21: the statement that CpGs are present at high level in the brain is simply wrong since each tissue has the same number of CpGs.
3) Line 38: the statement that 98 percent of methylation is on CpGs is in contrast with recent literature showing high non-CpG methylation show in the brain.
4) Line 74: it has never defined that CpG islands are 1000 bp long
5) Line 100: the statement on DNMTs should be rewritten since it doesn't reflect the message of the cited paper.
6) Line 126: the statement about DNMT3B needs to be referenced and explained.
7) Line 132: the statement is in contrast with a large body of literature showing that neurodegeneration is associated with hypomethylation.
8) Line 155 and 165: this is in contrat with the literature since for example old data on APP methylation were never confirmed and several papers showed differential methylation of PSEN1 (see for example the work of Fuso and colleagues).
9) The acronyms AD and PD already inlcude the "disease" so it is not necessary to add pathology.

Author Response

# Reviewer 3

The review by Kaur and colleagues is aimed at showing the role of the DNA methylation in neurodegenerative disorders and adds to a large body of similar papers. Unfortunately the manuscript results quite superficial and repetitive without drawing any relevant perspectives. The review needs of an extensive editing of both the English language and the style.

Response: Authors would like to appreciate reviewer’s comment for improvement in our manuscript. We have thoroughly revised the manuscript and suitable changes has been made (using track changes). The manuscript has now been revised grossly to provide valuable insight on DNA methylation in neurodegenerative diseases. Further this is to inform, that manuscript has now been revised to improve language and style of writing. The point wise response to the reviewer’s comment has been given herewith.

Response to specific comments:

  1. Line 19: better to use "epigenetic modifications" instead of "epigenetic alteration".

Response: As suggested the change has now been incorporated at the suitable place.

  1. Line 20-21: the statement that CpGs are present at high level in the brain is simply wrong since each tissue has the same number of CpGs.

Response: As suggested the sentence has been revised to remove any ambiguity.

  1. Line 38: the statement that 98 percent of methylation is on CpGs is in contrast with recent literature showing high non-CpG methylation show in the brain.

Response: As suggested the sentence has been corrected to remove any ambiguity.

  1. Line 74: it has never defined that CpG islands are 1000 bp long

Response: Correction in the CpGs DNS base pair length has been done.

  1. Line 100: the statement on DNMTs should be rewritten since it doesn't reflect the message of the cited paper.

Response: The ambiguity in the sentence has rectified by rewriting the sentence (section 4 first paragraph).

  1. Line 126: the statement about DNMT3B needs to be referenced and explained.

Response: The statement of DNMT3B has now been referenced and explained (section 4 last paragraph).

  1. Line 132: the statement is in contrast with a large body of literature showing that neurodegeneration is associated with hypomethylation.

Response: The ambiguous statement has now been removed for better clarity (section 4 last paragraph).

  1. Line 155 and 165: this is in contrat with the literature since for example old data on APP methylation were never confirmed and several papers showed differential methylation of PSEN1 (see for example the work of Fuso and colleagues).

Response: The sentence has now been simplified to remove ambiguity.

  1. The acronyms AD and PD already inlcude the "disease" so it is not necessary to add pathology.

Response: As suggested, term “pathology” has been removed after the AD/PD acronyms.

Round 2

Reviewer 3 Report

Both for the style and the superficiality of the text evidenced in the first review no significant changes were made despite the authors' claim in the response letter. The authors just added new text according to the other reviewers' requests but both all the old and the new text are superficial, confused and not clearly written. In particular the description of DNA methylation analyisis. To give an example more than one methods are identified as "gold standard", including old and no more applied methods. 

I cannot see any improvement of English language.

Finally, for five out of the nine specific points I have raised (specifically points 1, 5, 6, 7, 8), the authors claim corrections but the sentences are extactly as in the first version. This appears disrespectful and irratating. 

Author Response

Response to Reviewer’s comment

# Reviewer 3

Both for the style and the superficiality of the text evidenced in the first review no significant changes were made despite the authors' claim in the response letter. The authors just added new text according to the other reviewers' requests but both all the old and the new text are superficial, confused and not clearly written. In particular the description of DNA methylation analyisis. To give an example more than one methods are identified as "gold standard", including old and no more applied methods. 

I cannot see any improvement of English language. Finally, for five out of the nine specific points I have raised (specifically points 1, 5, 6, 7, 8), the authors claim corrections, but the sentences are extactly as in the first version. This appears disrespectful and irratating. 

Response: Author regret for the inconvenience caused. Further, the statement regarding “Gold standard” method has now been revised. To elaborate, the statement mentioning HPLC-UV technique for detection of dC and 5mC as Gold Standard test has been revised (Page 3 Line 136-137). The manuscript has now been thoroughly revised for the English language and grammatical errors for better clarity and understanding to the readers. These changes are shown in track changes.

We would like to highlight (in yellow) the previous changes incurred with your (point wise) suggestions.

Previous comments

The review by Kaur and colleagues is aimed at showing the role of the DNA methylation in neurodegenerative disorders and adds to a large body of similar papers. Unfortunately the manuscript results quite superficial and repetitive without drawing any relevant perspectives. The review needs of an extensive editing of both the English language and the style.

Response: Authors would like to appreciate reviewer’s comment for improvement in our manuscript. We have thoroughly revised the manuscript and suitable changes has been made (using track changes). The manuscript has now been revised grossly to provide valuable insight on DNA methylation in neurodegenerative diseases. Further this is to inform, that manuscript has now been revised to improve language and style of writing. The point wise response to the reviewer’s comment has been given herewith.

Response to specific comments:

  1. Line 19: better to use "epigenetic modifications" instead of "epigenetic alteration".

Response: Earlier authors though of specific change in the mentioned line is sought. However as per recent suggestion, “epigenetic alteration” has now been replaced with “Epigenetic modifications” throughout the manuscript and changes are shown in track change.

  1. Line 20-21: the statement that CpGs are present at high level in the brain is simply wrong since each tissue has the same number of CpGs.

Response: As suggested the sentence has been revised to remove any ambiguity.

  1. Line 38: the statement that 98 percent of methylation is on CpGs is in contrast with recent literature showing high non-CpG methylation show in the brain.

Response: As suggested the sentence has been corrected to remove any ambiguity.

  1. Line 74: it has never defined that CpG islands are 1000 bp long

Response: Correction in the CpGs DNS base pair length has been done.

  1. Line 100: the statement on DNMTs should be rewritten since it doesn't reflect the message of the cited paper.

Response: The ambiguity in the sentence has rectified by rewriting the sentence (section 4 first paragraph).

  1. Line 126: the statement about DNMT3B needs to be referenced and explained.

Response: The statement of DNMT3B has now been referenced and explained (section 4 last paragraph).

The statement regarding DNMT3B has now been elaborated and referenced and can be seen as track changes (Page 4-5, Line 189-195). The same has been given here as highlighted in yellow for quick reference.

“…. The DNMT family of enzyme includes DNMT1 and DNMT3A/DNMT3B enzymes. DNMT1 preferentially leads to methylation of hemi-methylated DNA and further main-tains DNA methylation after replication of DNA. Whilst DNMT3A and DNMT3B causes methylation on non-methylated and hemi-methylated DNA equally and are considered as de novo methyltransferase. DNMTs go through significant conformational changes, are capable of oligomerization, and can self-inhibit, all of which could help to regulate their activity [36].”

  1. Line 132: the statement is in contrast with a large body of literature showing that neurodegeneration is associated with hypomethylation.

Response: Authors has now provided more comprehensive overview of altered methylation status considering both hyper and hypomethylation status. The changes have now been incorporated in Section 5.1 (Page 8-9; Line 268-275) and can been seen in track changes. For quick reference, the same has been provided herewith (highlighted in yellow).

“…Decreased global DNMT1 and 5mC in temporal cortex and reduced global 5mC and 5hmC in hippocampus of AD patients has been reported [80]. In contrast, some study supports elevated level of 5mC and 5hmC in frontal, temporal cortex, and hippocampus in AD [81,82]. Further, reduced 5mC level in APP, PSEN1, SERT1 promotor in brain and blood of AD patients has been recorded [83,84]. Further a recent study supported a posi-tive correlation between 5mC and 5hmC value in AD patients [85]. This could be helpful in differential diagnosis of AD patients from PD (with lower 5hmC level and unchanged DNMT3A expression).”

  1. Line 155 and 165: this is in contrat with the literature since for example old data on APP methylation were never confirmed and several papers showed differential methylation of PSEN1 (see for example the work of Fuso and colleagues).

Response: The contrasting data for APP and PSEN1 has now been revised to add more comprehensive information of APP and PSEN status in AD patients. The changes can be seen in track changes (Page: 9-10; Line: 290-309). For quick consideration the added paragraph has been give below (highlighted in yellow).

“.. Important variations in DNA methylation profiles of the APP, microtu-bule-associated protein tau, and GSK3B were also discovered, but not of PSEN1, be-ta-secretase 1 precursor, or apolipoprotein E. Many studies have reported genetic variants linked to enhanced AD susceptibility, but do not limit many other promoters. Almost 28 gene locations are linked with AD, while little is known about bridging integrator- 1 func-tion in AD pathogenesis, it may have considerable impact on tau pathology, amyloid precursor protein endocytosis, and inflammation in neurons [88-91].

Most of the AD studies performed till date have mostly considered a gene-directed analysis, thus, the methylation of promoter genes of AD, (especially APP) has been widely explored. As mutant APP gene poses a risk factor for AD, it is apparent that epigenetic modifications of APP promoter leading to enhanced gene expression are also risk factors for AD. West and colleagues reported hypomethylation of the APP promoter in AD patient [86]. Further, some contrasting study rejected these findings suggesting it as an outcome of larger epigenetic modification rather than specific alteration in methylation [83].

Further, expression of cortical PSEN1 appeared stable during embryonic develop-ment while upregulated levels have been reported in AD patients. The cortical PSEN1 ex-pression in rodent appears tightly regulated by histone modification at various stages of neuronal development. Generally, PSEN1 gene remains partly methylated and sup-pressed after development, while hypomethylation of PSEN1 is reported associated with its elevated expression in AD population [87].”

  1. The acronyms AD and PD already inlcude the "disease" so it is not necessary to add pathology.

Response: As suggested, term “pathology” has been removed after the AD/PD acronyms.

Round 3

Reviewer 3 Report

I can appreciate an improvement in style but some things are still to be reviewed: 

Line 55-58: the statement that methylation on DNA takes place in a non-CpG dinucleotide frame-work doesn't make any sense. In addition, most of the test is focused only on non-CpG methylationand this makes it incomplete.

Line 283-288: the fundamental concept of the period is unclear. In the first sentence the statement that outlines an absence of variation of the methylation profile of the PSEN1 promoter is wrong and is opposite to the last and correct sentence which outlines its variation. I recommend a better study of the literature about it.

Author Response

Response to Reviewer’s comment

# Reviewer 3

I can appreciate an improvement in style but some things are still to be reviewed: 

Response: Author would like to thank reviewer for providing valuable insights for the improvement in the manuscript.

Query 1: Line 55-58: the statement that methylation on DNA takes place in a non-CpG dinucleotide frame-work doesn't make any sense. In addition, most of the test is focused only on non-CpG methylation and this makes it incomplete.

Response: The sentence regarding methylation has now been revised as following.

“In general, methylation of DNA is found at majorly at CpG dinucleotide, however In in somatic cells, methylation on DNA is highly prevalent takes place in aon non-CpG dinu-cleotide framework.”

Reference: Ramsahoye BH, Biniszkiewicz D, Lyko F, Clark V, Bird AP, Jaenisch R. Non-CpG methylation is prevalent in embryonic stem cells and may be mediated by DNA methyltransferase 3a. Proc Natl Acad Sci U S A. 2000 May 9;97(10):5237-42. doi: 10.1073/pnas.97.10.5237. PMID: 10805783; PMCID: PMC25812.

Regarding DNA methylation test on CpG has already been discussed in section 3, last paragraph (line 169-179), which is highlighted as follows:

“Luminometric methylation assay is a technique which combines two DNA restriction di-gest operations that are run in parallel, followed by pyrosequencing to fill in the gaps be-tween the digested DNA strands' projecting ends. The CpG-methylation-sensitive enzyme HpaII is used in one digesting step, whereas the methylation-insensitive enzyme MspI is used in the other, cutting at all CCGG sites [29]. Bisulfite sequencing is the "gold standard" technology in DNA methylation research. Recent DNA sequencing technology can't tell the difference between methyl-cytosine and cytosine. The deamination of cytosine into uracil is mediated by bisulfite treatment of DNA, and these transformed residues are read as thymine by PCR amplification and subsequent Sanger sequencing analysis. 5 methyl-ated Cytosine residues, on the other hand, are independent to this change and will remain as cytosine. By comparing the” Sanger sequencing” reads from DNA samples which that remain untreated to the cloned sample after bisulfite treatment, the methylated cyto-sines5mC can be detected. This approach may now be expanded to DNA methylation analysis over a complete genome because of the advent of next-generation sequencing (NGS) technology [30].”

Query 2: Line 283-288: the fundamental concept of the period is unclear. In the first sentence the statement that outlines an absence of variation of the methylation profile of the PSEN1 promoter is wrong and is opposite to the last and correct sentence which outlines its variation. I recommend a better study of the literature about it.

Response: The first sentence has now been revised and corrected. The corrected sentence is as follows:

“Important variations in DNA methylation profiles of the APP, microtu-bule-associated protein tau, and GSK3B were also discovered, along with PSEN1, beta-secretase 1 precursor, or apolipoprotein E.”

This is now in coordination with recent literature.
